# Multichannel Acoustic Spectroscopy of the Human Body for Inviolable Biometric Authentication

**DOI:** 10.3390/bios12090700

**Published:** 2022-08-31

**Authors:** Hyung Wook Noh, Chang-Geun Ahn, Seung-Hoon Chae, Yunseo Ku, Joo Yong Sim

**Affiliations:** 1Bio-Medical IT Convergence Research Department, Electronics and Telecommunications Research Institute, Daejeon 34129, Korea; 2Department of Biomedical Engineering, College of Medicine, Chungnam National University, Daejeon 35015, Korea; 3Department of Mechanical Systems Engineering, Sookmyung Women’s University, Seoul 04310, Korea

**Keywords:** acoustics, biometrics, anti-spoofing, access control, multisensor systems, spectral analysis, human–machine interactions

## Abstract

Specific features of the human body, such as fingerprint, iris, and face, are extensively used in biometric authentication. Conversely, the internal structure and material features of the body have not been explored extensively in biometrics. Bioacoustics technology is suitable for extracting information about the internal structure and biological and material characteristics of the human body. Herein, we report a biometric authentication method that enables multichannel bioacoustic signal acquisition with a systematic approach to study the effects of selectively distilled frequency features, increasing the number of sensing channels with respect to multiple fingers. The accuracy of identity recognition according to the number of sensing channels and the number of selectively chosen frequency features was evaluated using exhaustive combination searches and forward-feature selection. The technique was applied to test the accuracy of machine learning classification using 5,232 datasets from 54 subjects. By optimizing the scanning frequency and sensing channels, our method achieved an accuracy of 99.62%, which is comparable to existing biometric methods. Overall, the proposed biometric method not only provides an unbreakable, inviolable biometric but also can be applied anywhere in the body and can substantially broaden the use of biometrics by enabling continuous identity recognition on various body parts for biometric identity authentication.

## 1. Introduction

Biometrics is a technology that uses the physical and behavioral characteristics of the human body to identify individuals and to give them access to systems [1]. Biometric authentication verifies personal identities without requiring users to remember passwords or carry physical objects. Physical objects mainly used in the field of traditional biometric authentication include, for example, personal ID cards, magnetic cards, passports, and keys. These physical entities must always be present to properly identify an individual, and there is always a risk of theft or loss. Thus, biometrics can be used to increase user convenience for cyber transactions, online and physical access, seamless interfacing with internet-of-things (IoT) devices, and raise the levels of personal, corporate, and national security [2]. Fingerprint, iris, and facial recognitions are most commonly used in biometrics, as they provide physical characteristics that are unique to each individual. However, one can make fake copies of these biometric markers for spoofing because they rely on the structural features of the acquired image [3,4]. In particular, new fingerprints, faces, or irises cannot be re-issued to users as new passwords can. Therefore, once a template is stolen, it becomes a permanent threat for spoofing. Furthermore, stealing this template is considered as a breach of customer or employee privacy and represents the improper disclosure of sensitive information. Finger vein technology has been reported to have advantages over previous technologies, such as reduced risk of theft and inaccuracies due to skin defects [5]. A. H. Mohsin et al. conducted a review of current technologies in vein biometric authentication and referred that this technology is more resistant to counterfeiting and duplication than other types of biometrics because it uses the geometric information of blood vessels inside the skin [6]. Önsen Toygar et al. have also conducted research on creating anti-counterfeiting multimodal authentication systems using three biometric traits of vein: palmar, dorsal and wrist vein [7]. Another recent study introduced a unique biometric recognition technique based on vein patterns on the back of the hand obtained from thermal imaging using a vein identification system (VPID) [8]. However, it has been reported in several studies that it is possible to attack the finger vein recognition system using fake finger vein images. If a vein image captured by a near-infrared imaging device is leaked, it can be printed and successfully used for spoofing attacks against the vein recognition sensor [5,9]. To overcome these challenges, researchers have been developing techniques to detect spoofing attacks, such as liveness detection [10], devising new biometric modalities [11,12], and combining two or more biometric modalities to increase the level of security [13,14].

Different people perceive varying frequencies of sound differently [15,16] because the cochlear structures and biomaterial characteristics in the ear are different in different people. This fact led us to explore and utilize the anatomical and biomechanical features of the human body as identity recognition signatures to distinguish different people. The data acquired by the bioacoustic measurement system are in spectral forms and exhibit different features at each frequency. In this case, each frequency exhibits high local correlations with nearby frequencies and decreasing local correlations with frequencies farther away. Therefore, using every frequency feature can be inefficient. Certain frequency features could act as noise, and redundant frequency data can lower the performance because of the curse of dimensionality [17,18]. For this reason, feature selection is essential for optimizing frequency–domain bioacoustic systems. In addition, selecting a subset of relevant features through feature selection can achieve a faster operation time as well as enhanced generalization by reducing overfitting.

In this work, to achieve an accuracy comparable to that of state-of-the-art fingerprint and iris biometrics, we designed a multichannel identity authentication system based on bioacoustic spectroscopy and evaluated how the accuracy changes as a function of the number of sensing channels. In addition to the number of receiving channels, we investigated individual frequency features by comparing the performances at different numbers of selected frequencies to the performance obtained using the full-frequency spectra in conjunction with the use of forward-feature selection and an exhaustive combination search. The performance of a convolutional neural network (CNN) algorithm that automatically extracts hierarchical features is compared with those achieved by other machine learning algorithms based on feature selection.

Our previous study reported the feasibility of bioacoustics as a biometrics technology using a single bioacoustic finger channel [19]. The performance, however, was insufficient (accuracy, 95.89% and operation speed, 15 s) compared with the long-established biometrics such as fingerprint, iris, etc. Commercial biometric authentication systems perform authentication in just a few seconds [20] and are highly accurate in the case of fingerprints (less than 1% of EER [21]). Therefore, we performed this study with the goal of achieving high accuracy and fast recognition speed comparable to commercial biometric recognition system. Furthermore, we provide guidance of the system design trade-off for application to various practical acoustic biometric authentication fields.

## 2. Related Work

### 2.1. Liveness Detection Techniques

Liveness detection techniques that have been implemented include time-series analysis [22], material properties analysis [23], data-driven machine learning [24,25], and optical reflection analysis [26,27]. However, these methods are still at risk of spoofing because these characteristics can be mimicked [4]. In addition, studies have been conducted to simultaneously perform both liveness detection and identification using electroencephalograms (EEGs), electrocardiograms (ECGs), and bioimpedance signal patterns [28,29,30,31] that can even overcome the limitations of the image-based features of fingerprints or iris. The liveness detection methods based on EEG and ECG are useful because it is difficult to reproduce these signals and make fake copies. However, these methods are not robust enough for independent identification because the signals can change depending on the emotional or physiological states of the user [32]. Our proposed method tackles this problem by extracting the internal anatomical and structural information of the human body, which is more robust against changes of the physiological and emotional states.

### 2.2. Acoustic System Identification and Imaging

Similar to structural system identification and ultrasound imaging methods, bioacoustics has been applied to the human body to extract and distinguish a subject’s dynamic frequency response. Our previously reported study used a single bioacoustic finger channel by employing a transducer and an adjacent sensor to measure the transmitted bioacoustic signals that reflected the anatomical and biomaterial properties of the human body. However, this identity recognition method was less accurate than state-of-the-art fingerprint and iris recognition systems [21]. Nevertheless, there is a potential to increase the accuracy of bioacoustics by acquiring more information from additional channels at different locations on the body for both recognition and liveness detection.

### 2.3. Bioacoustics for Biomechanical Characterization

In this regard, bioacoustics is an ideal method for obtaining the biomechanical characteristics of the human body. For instance, the sound from touching or tapping the skin of the hand and arm has been used as a gesture input for interacting with smart wearable devices [33,34,35] that are equipped with ultrasound transducers or accelerometers. In addition, breathing sounds [36] and shared secret knocks [37] have been employed to demonstrate user authentication. Bioacoustic sensing has also been used to detect the grasping of objects based on the sound generated by the touched object and propagated through the hand and arm [38]. Structural system identification [19] is a technology that is closely related to the employment of acoustic signals for detecting the internal structural and material characteristics of structures. This method is used to noninvasively monitor structures and detect damage by precisely acquiring and analyzing the vibration modes and dynamic frequency response of the mechanical structure and by detecting changes caused by internal structural defects and aging [39,40]. Ultrasound imaging is another representative example that demonstrates the use of acoustic signals to analyze the internal structural features of the body. Ultrasound imaging extracts the structural information of the internal organs by analyzing the echo signals reflected back to the transducer [41].

## 3. Multichannel Bioacoustic Identity Authentication

### 3.1. Multichannel Bioacoustic Identity Authentication System

The human body can be considered a musical instrument that has a unique shape and material composition [42,43]. The proposed method evaluates these human body traits by vibrating a body part, specifically the hand, and listening to the propagating sound as the excitation frequency is altered. Conventional biometrics compares the structural similarities between the acquired images (Figure 1a), whereas the proposed biometric authentication method uses the internal characteristics of the body and extracts the information in the frequency domain that creates a new higher level of biometric security. Assuming the existence of a system that consists of numerous masses, springs, and dampers (Figure 1b), the proposed method applies specific vibrations to these mass–spring–damper systems to extract their bioacoustic transmittance characteristics and uses these characteristics to distinguish people. For example, as shown in Figure 1b) (finger cross-section v–v’), each bone corresponds to an object that transmits vibrations that contribute to the dynamic response, as do all the joints that connect the bones and other tendons and ligaments surrounding the hand. As shown in the cross-section w–w’ of Figure 1b, each bone is surrounded by fatty tissues, muscle, and skin that act as dampers that determine the dynamic response characteristics.

As shown in Figure 2a, the proposed system utilizes a transducer (3W, Sparkfun Inc., Niwot, CO, USA) to generate a vibration and five sensing microphones (SPU0410LR5H, Knowles, IL, USA) to receive signals simultaneously. Waveform generation and frequency selection are performed by a waveform generator (AD9833, Analog Devices, MA, USA). The AD9833 is a low-power, programmable waveform generator capable of producing sine wave outputs. The excitation frequency can be modulated by software programming and can be easily tuned without external components. The generator operates by 3-wire serial peripheral interface (SPI) communication with the MCU. Specifically, a frequency selection command is sent from the computer to a microcontroller unit (MCU, Arduino Nano, Italy), and the required frequency is sent to AD9833 via SPI. The generated waveform controls the bone conduction transducer via a transducer driver. The transmitted signal is simultaneously received by each microphone through an independent phase-sensitive lock-in amplifier configured with an analog demodulator (AD630, Analog Devices) and a low-pass filter. Lock-in amplifiers can sensitively extract signals of interest, even in the presence of significant noise [44].

Therefore, to extract a signal corresponding to a specific frequency from the region of interest, a homodyne detector composed of a modulator and a demodulator followed by low-pass filters with a cut-off frequency of 20 Hz, which is sufficient to reject the AC signal from the excitation, is used as shown in Figure 2, varying the excitation frequency for phase-sensitive detection. The dc level of the output (obtained by low pass filtering) is proportional to the amplitude of the excitation signal and the phase difference between the signal received by the sensor. Therefore, the better the signal transmission at a specific frequency, the larger the output signal. Then, the output signal is analog-to-digital converted (ADC) with a sample frequency of 18.8 Hz by the MCU and then sent to the computer. This is achieved via a custom-built code in MATLAB (R2019b, MathWorks, Natick, MA, USA) that also stores the data. The data acquisition unit was controlled by communication between MCU and a computer and interfaced with a user identification module. The computational module was comprised of user interface, data storage, and classification algorithm. The classification models were trained and tested to distinguish one from the others in closed-set scenarios. Figure 2b shows a photo of the system in use, and the insets show the microphones and a vibrational transducer. One of the advantages of this configuration is that the transducer can simultaneously emit signals to multiple sensors, that is, the single-input–multi-output configuration is possible [45,46]. In the current configuration, the addition of a sensing channel allows the simultaneous acquisition of various patterns of signals without weakening the signal strength. Additionally, each channel is equipped with an independent frequency demodulator that selectively extracts and stores the signal at each frequency, thereby minimizing external noise and improving reproducibility. A thimble design in the measurement setup shown in Figure 2b minimizes changes in finger positioning for every measurement, thus improving reproducibility. Figure 2c,d shows transmitted and received signals with and without a hand on the system, respectively. The microphone is covered with a polydimethylsiloxane (PDMS) layer as an impedance matching layer to achieve a large signal-to-noise ratio. Appendix A shows that the signal received by a microphone covered with an impedance matching layer is increased almost ten times compared with an uncovered microphone. The experiment was conducted by modulating the frequency from 200 Hz to 3 kHz at 20 Hz intervals. Therefore, in total, 141 acoustic impedance values were obtained for each finger, and 705 feature vectors were used directly as inputs to the classifier algorithms. Fifty-four human subjects underwent clinical trials. The subjects participated in three to five independent sessions over 4 months and provided 20 measurements per day. The measurement data were used to evaluate the identification performance of bioacoustic spectroscopy. To characterize the acoustic spectrum of different fingers, all of the five fingers (thumb, index, middle, ring, and little) were tested simultaneously. For each finger measurement, five sensors were placed on the center of each fingertip. The transducer was positioned between the palm and carpal bones near the wrist (Figure 2b), allowing the signal to be transmitted from the transducer to each of the five sensors through each finger bone. The transmitted acoustic signal contains anatomical information about the fingers (bone, cartilage, tendon, and muscle tissues) and relies on their geometries as well as their biomaterials and biomechanical properties [47,48]. Principal component analysis (PCA) of the acoustic spectrum of each finger was conducted using MATLAB.

### 3.2. Multichannel Biodynamic Response

The differences in the shape, length, and the anatomical structure of each finger are expected to produce different acoustic transmission characteristics for each finger. The acoustic transmission spectra of each finger (thumb, index, middle, ring, and little) were measured, and various discriminative machine learning models were tested to verify if the acoustic spectrum for each finger is distinguishable. The acoustic transmitted signal was acquired for one subject 20 times per day, and data were collected on three independent days. The measured signals of each finger varied considerably with frequency, as shown in Figure 3a, and because the magnitude and slope of the signals for each finger were different, fingers could be visually distinguished using the spectra. PCA [49] was conducted on the acquired data from all five fingers to visualize the distinguishability of the acoustic transmission signal from each finger. The first three principal components of the spectra are displayed in Figure 3b and are clearly grouped for each finger, thus enabling finger classification. To classify the five fingers, discriminative classification models were utilized. Multiple machine learning algorithms of k-nearest neighbor (kNN), linear support vector machine (SVM), random forest (RF), and linear discriminant analysis (LDA) in Figure 3c were tested because these machine learning models were previously reported as baseline models [19].

The experimental results indicate that the discernible spectral differences between the fingers can be used as biometric features. This result agrees with that of a previously reported bioacoustics identity recognition system. Note that the measurement of each finger channel was implemented simultaneously in contrast with the previous work, wherein the measurement was repeated by changing the finger on the same device. For visualization purposes, the graph is plotted from 200 Hz to 1 kHz, and the full-spectral pattern (200 Hz to 3 kHz) is shown in Appendix A.

### 3.3. Interpersonal Variation of Multichannel Biodynamic Response

In order to test the interpersonal variation of the acoustic signal transmission characteristics for each finger channel, the acoustic transmitted signal was acquired for five subject 20 times per day, and data were collected on three independent days. The signals from each finger channel were recorded from five subjects to determine the differences in the signal transmission characteristics of each person. Acoustic transmittance data for each finger channel are shown in Figure 4a and Appendix A (full-spectral pattern from 200 Hz to 3 kHz). For each finger channel, spectral pattern data from all five subjects exhibit differences in both the slope and magnitude. The spectra from each subject have unique patterns and slopes characteristic for each finger channel that show clear differences among subjects. Similar patterns can be observed in certain finger channels and frequencies among different subjects, but each subject is clearly different in another channel and frequency that can eventually increase the accuracy of interpersonal classification. For instance, the trends in the frequency range at around 400 Hz measured from the middle fingers of subjects 3 and 4 appear to be similar but are very clearly distinguished in other finger channels (e.g., thumb and ring fingers). To visualize the distinguishability of the acoustic spectra from each subject, PCA was conducted on the data obtained from five different subjects using merged channel data from each of the subjects’ five fingers, and the first three principal components of the spectra are displayed in Figure 4b. The acoustic spectrum for each subject is distinctly clustered, proving the clear distinguishability of different subjects. To verify the temporal stability of the acoustic transmission signal, the finger acoustic spectra of the five participants were acquired on three different days at two-week intervals.

### 3.4. Temporal Changes of Multichannel Biodynamic Response

As shown in Figure 5a, the interpersonal differences of the finger acoustic transmission spectra were larger than the four-week temporal changes (i.e., after two two-week intervals). The intensity images for the bioacoustic signal are shown in Figure 5b, wherein different channels are concatenated across the width of the two-dimensional image to qualitatively visualize the intrapersonal similarities and differences. The measured spectra of the subjects had different morphological shapes, patterns, and slopes, and varied considerably depending on frequency. These results imply that the acoustic transmission signals provide interpersonal differences as well as excellent temporal stability. Thus, the acoustic transmission spectrum has promising potential as a biometric trait and can provide excellent permanence and uniqueness. In the case of iris recognition, Das et al. [50] and Johnson et al. [51] performed longitudinal studies for up to 1.5 years and 3 years, respectively, and found that a statistically significant loss of accuracy occurred in a period of 3 years, but it was not at a level that prevented practical use. In the case of face recognition, it is known that it causes a greater reduction in accuracy than iris. For instance, Deb et al. [52] reported that the true acceptance rate at 0.1% false acceptance rate decreased from 81.94% to 49.33% when the interval was extended from one year to three years for children. In our previous study [19], we evaluated the permanence of the bioacoustics features of a finger over two months and found that the features maintained at least over two months. In this present work, we assessed the discriminative power over four months and found that the features of each finger were preserved and still discriminative among subjects. Since our methods were evaluated from 2 months to 4 months, it is necessary to additionally conduct longer longitudinal studies in future work to evaluate characteristic changes over several years as such existing longitudinal studies on face and iris recognition.

## 4. Biometric Authentication Using Multichannel Bioacoustic Signals

### 4.1. Machine Learning Algorithms

For the performance evaluations, several classifiers (RF, kNN, LDA, and SVM) were configured in MATLAB with an error-correcting output code [19]. The kNN classifier that yielded the highest accuracy in our other previous study [31] was also tested. The kNN classifier estimates the class label of a new observation by using the majority class of the k-nearest neighbors [31]. The nearest neighbor number was set to 1 after a parametric search from 1 to 20, and the Euclidean distance metric was used. The SVM was also tested, the second polynomial kernel function was used, and the tuning parameter was set to 1.0 [53]. The RF consisted of 705 trees, and the Gini diversity index was used. Other features were set to the default setting, and LDA was configured using the default settings. Using the threshold values of the latent variables of the classifier, the measured spectral data were assigned a subject class only when the data had a posterior probability larger than the threshold. The performance of each discriminant model can be evaluated based on closed-set identification so that it can assign a given observation to the certain class with the highest latent variable that can distinguish it from others. If the feature vectors obtained from biometrics do not give a sufficient probability, the system should be able to reject it. Using the classifier’s latent threshold, the measured data was assigned to a class only if the posterior probability is larger than the threshold. In addition, the sensitivity (1–false acceptance rate, FAR) and specificity (1–false rejection rate) were assessed by changing the thresholds of posterior probabilities. Then, AUC and EER were also obtained and evaluated from the receiver operating characteristic (ROC) curve. To verify the classification accuracy, each classifier was evaluated using five-fold cross-validation on 5232 datasets from 54 subjects. The data from one subject on the same date tend to be more similar than data measured on another date. The effects of the measurement environmental factors such as posture, temperature, and humidity may vary from day to day. Thus, if the data measured on the same date are included in both the train-set and the test-set, the test data become easier to predict. However, in reality, we should predict data on a day different from the measurement date of the data used in the model. Therefore, to mutually exclude each subject’s data measured on the same date, each subject’s data were divided into five groups by indexing data from 1 to 5 according to the measurement date. Through this process, the original dataset was partitioned into five almost equal-sized subgroups, and five-fold cross-validation was performed on each group of data.

### 4.2. Biometric Authentication with Increasing Finger Channels

To investigate the effect of finger channel selection on performance, the accuracy of each possible finger channel combination was compared using the classification models. The highest classification accuracy of 99.08% was achieved with the RF classifier, and all models yielded accuracies > 90% as shown in Figure 6. To compare the identity recognition performance of different numbers of finger channels, the accuracy corresponding to each channel combination was evaluated using the mean and standard deviation, as shown in Figure 6a. The classification performance using the RF classifier showed an average performance of 92.61%, even if only two finger channels were used. The accuracy initially increased from 79.17% to 92.61% as the number of channels increased from one to two, and it further increased to 96.72%, 98.26%, and 99.08% for three, four, and five channels, respectively. Overall, the multichannel bioacoustic identity authentication system increased the classification accuracy of 54 subjects from 79.17% to 99.08% and reduced the classification error rate from 20.83% to 0.92% (a factor of 22). Notably, the increase in the classification accuracy decelerated as the number of finger channels increased. This trend indicates that there is a trade-off between the additions of more channels and the increase in accuracy. Figure 6b shows the accuracy of each finger channel combination in the RF model that achieved the highest accuracy among the four classifier models. We assigned numbers 1 to 5 from the thumb to the little finger (in this order), and these numbers were marked on each bar to indicate the combined finger channels. As expected, the accuracy gradually increased with an increasing number of finger channels. This emphasizes the trade-off between system complexity, i.e., the number of channels used and accuracy.

Moreover, different channel combinations with the same number of fingers yielded mostly similar performances, thus indicating that the accuracy of the system is independent of the selected fingers. The detailed results of Figure 6a,b are shown in Appendix A, respectively.

### 4.3. Frequency Feature Selection

In addition to the comparison of the accuracy of the finger channel combinations, the changes in accuracy for different acoustic frequency selections were also analyzed. The measurement frequency ranged from 0.2 kHz to 3 kHz in all 54 subjects. This range was divided into ten frequency sections, and the classification accuracies with all possible combinations of frequency sections were evaluated by selecting three out of ten sections (i.e., 120 combinations). The accuracy of each selected frequency combination is shown as an intensity image map in Figure 7a, wherein higher accuracy combinations are presented at the upper position of the intensity map, whereas unselected frequencies are marked in blue. The combination of the three selected frequency sections of 0.2–0.3 kHz, 2.1–2.4 kHz and 2.7–3.0 kHz resulted in the highest classification accuracy of 98.93% as located at the top of the intensity map. This row-wise sequential rank of classification accuracies in Figure 7a suggests that simultaneously selecting the low- and high-frequency sections is advantageous for achieving a higher accuracy. The middle-frequency ranges were related to increased accuracy to a lower extent compared with the low- and high-frequency sections. This result indicates that the frequencies’ regions below 0.6 kHz and above 1.8 kHz are salient regions for classification. To illustrate the relative importance of the frequency sections to the classification accuracy when using three frequency sections, we evaluated the average classification accuracy of the predictions from all combination cases (Figure 7b). As expected, the mean accuracy based on the frequency sections from 0.6 to 1.2 kHz decreased before they increased from 1.2 to 3 kHz. The forward-selection method was used to test how the increase in the number of selected frequencies affects the classification accuracy [54]. This method increases the number of frequencies one-by-one by selecting the frequency that has the most distinctive performance from 141 measured frequencies (Figure 7c). The detailed performance values of Figure 7b,c are represented in Appendix A, respectively. Remarkably, the highest accuracy was achieved by using only one-third of the frequency range.

The accuracy exceeded 90% when five frequencies were included, and the accuracy using 15 frequencies was almost the same as that of using all 141 frequencies. The highest performance of 99.62% was achieved when 47 to 83 frequencies were selected (except the 99.54% accuracy when 71 were frequencies selected), which was higher than that when all frequencies were used. Specifically, in addition to the RF method, the other classifiers (kNN, LDA, and SVM) also performed best when 47 selected frequency features were used compared with the case at which other numbers of frequencies were used (Table 1). Using the frequency optimization method and including only 47 frequency features reduces the measurement time by one third, requiring only 2.5 s. These results indicate that selecting the appropriate frequencies can reliably increase system performance. Accordingly, the use of a small subset of the spectrum can reduce the acquisition time while also outperforming the full-frequency scan.

### 4.4. Deep Learning Implementation of Multichannel Bioacoustic Identity Authentication

To analyze the predictions for acoustic signal classification based on frequency selection, we applied a modified CNN algorithm proposed by Salamon and Bello (SB-CNN) [55]. Deep CNN is a multilayered architecture that creates categories by progressively learning higher-level features on a layer-by-layer basis [56]. A CNN is a type of artificial neural network that is usually designed to extract features and classify high-dimensional data [57]. A typical CNN includes an input and an output layer, as well as the number of convolutional and subsampling layers optionally followed by fully connected layers [58]. The SB-CNN has three convolutional layers, two max-pooling layers, and two fully connected layers. The CNN is specifically designed for 2D time–frequency patch data. In our study, the transmitted acoustic signals of each channel were one-dimensional vector signals of each finger that consisted of 141 samples, and the five-finger channel data were reshaped into 2D (141 × 5) inputs. Figure 8 illustrates the detailed architecture of the modified SB-CNN with the reshaped input shape. The first two convolutional layers were followed by max-pooling with a stride size equal to the pooling dimensions, reducing the dimension of the spatial maps. The model was trained on 2000 epochs with an Adam optimizer [59], a constant learning rate of 0.0001, a mini-batch size of 128, and a dropout of probability 0.5 between dense layers [60]. We thus performed CNN training on these four cases with selected frequencies of 5, 15, 47, and 141. The detailed input data for the altered number of frequencies of the CNN are in Appendix A. The ROC curves for the classifier models for each case of selected frequencies are shown in Figure 9. LDA and RF classifiers outperformed *k*NN and SVM in all frequency selection cases.

When fewer frequencies were selected, SB-CNN demonstrated relatively poorer performance than other machine learning models. However, as more frequencies were selected, the accuracy increased to values that were almost equal to that of the RF classifier. In the RF model, the highest accuracy was obtained when 47 frequencies were selected by forward selection, whereas in the CNN model, as the number of frequencies used increased, the accuracy increased. We speculate that the main reason for these results is that deep-learning-based networks incrementally learn complex and abstract, hierarchical high-level features from data as opposed to machine learning models that require manual feature extraction [61]. Nevertheless, the accuracy of the RF classifier with 47 forward-selected frequencies was 99.62% (the highest overall). The accuracy, EER, and AUC for the tested classifiers are listed in Table 1.

When 47 optimal frequencies were selected, the RF model achieved the highest classification accuracy of 99.62% and the lowest EER of 0.0887%, whereas when all 141 frequencies were used, the SB-CNN classifier produced a better AUC value. The SB-CNN has an output layer composed of 54 soft-max nodes corresponding to the subjects enrolled in the dataset, which performs closed-set identification. For closed-set identification, the cumulative matching characteristic (CMC) curve is commonly used [62,63,64,65]. Whereas the ROC curve represents the quality of a 1:1 matcher, the CMC curve judges the ranking capabilities of the identification system [66]. Therefore, we further validated our system using the CMC curve to clearly evaluate the performance of closed-set identification. The CMC curves for the classifier models for each case of selected frequencies are shown in Figure 10a. The horizontal axis represents the number of top ranks considered, and the vertical axis represents the probability of the correct identity within the considered top ranks. The identification rate is an estimate of the probability that the subject is correctly identified, at least at rank-k. Test data is given rank-k when the actual subject is ranked in position k by the identification system. The area above the CMC curve effected by the number of subjects is considered as a percentage of the total number of subjects. Figure 10b shows the confusion matrix of 54 subjects in the RF model in which 47 forward-selected frequencies were used as input features. The blue boxes represent accurate predictions with the increasing color intensity representing higher accuracy, while the red boxes represent false predictions. Most of the predictions are located in the blue-colored diagonal matrix. As shown in Figure 10a, the curves depended on the number of the selected frequency as well as the classification models. The biometric features that showed better results in ROC produced a lower area above the CMC curve. For example, when the number of frequencies is 5, the LDA model shows the best performance in the ROC and is also the closest to the upper left corner in the CMC curve. In addition, when 47 frequencies are selected, the RF model, which shows the best performance in the ROC curve, also has the highest identification rate for all ranks in the CMC curve. The results show that the ROC curve and CMC curve are highly correlated as performance indicators.

### 4.5. Computational Time for Classifiers

Our machine learning models’ training took 2.11 s for RF, 0.10 s for kNN, 0.26 s for LDA, and 17.41 s for SVM, which took the longest time for training among machine learning models. In the SB-CNN, it took 1.23 s for 1 epoch training, and the total training time of 2000 epochs was 41 min. SB-CNN required longer training time to achieve optimal performance compared with machine learning models. However, in the case of machine learning, for feature optimization, preprocessing such as frequency selection may be necessary to improve performance, which may take additional time. However, since training in deep learning models such as CNN automatically learns well-distilled features, better models can be trained without additional preprocessing. The prediction time listed in Table 2 shows how fast each classifier can process input data and yield authentication results.

It is crucial that the model has a short enough prediction time in order to perform real-time authentication. The time required for prediction in each classification model was all within 212 ms, and the shortest prediction time (0.16 ms) was required in the deep learning model, SB-CNN.

### 4.6. Scalability Analysis

Although the accuracy obtained by our proposed system was comparable to state-of-the-art biometric technologies, we further investigated the scalability of our system to larger population. We analyzed the variation in classification accuracy with changes in the number of subjects and obtained an accuracy prediction curve based on experimental data. We changed the number of subjects by five intervals, and each dataset was randomly selected from a total of 54 subjects. Each classification accuracy was extracted using the RF model from each selected dataset.

These processes were repeated 10 times to extract average accuracy and standard deviation. In existing biometric authentication methods, performance may decrease linearly as the number of users increases [67,68], or the rate of decrease in performance can be decelerated non-linearly [69]. As shown in Figure 11, the results of the analysis were presented as boxplots, and the accuracy was shown to decrease as the number of selected subjects increased. We note that the decrease in the classification accuracy was decelerated with the increasing number of subjects. The slowdown of the accuracy degeneration led us to assume that the accuracy prediction curve as the number of subjects increased is non-linear and can be extrapolated by fitting a log-scale exponential function to the data.

As shown in the change in the slope of the prediction curve, the curve converged to 98.96%, and these results provide evidence that our system has the potential to provide the accuracy to be used for a practical biometric authentication application for over thousands of individuals. To support further, we additionally analyzed the variation in signal amplitude for each frequency for intra-class versus inter-classes in order to statistically compare the effect of each acoustic spectrum, as shown in Appendix A. Intra-class variability was expressed as the mean and standard deviation of all subjects after calculating the standard deviation for each frequency of the data within each class. In the case of inter-class variability, the signal amplitude within each class was averaged for each frequency, and the mean and standard deviation were calculated for all classes. To investigate statistical differences between classes, we divided the inter-class variation by the intra-class variation to obtain a quantitative pattern for each frequency, which is shown in Appendix A. The largest difference between classes is in the low-frequency region, followed by a slightly higher inter-class difference in the high-frequency than in the mid-frequency region, which is highly correlated with the results shown in Figure 7. In future work, we will validate our method on a large population to provide direct evidence that the method can perform on a large-scale with accuracy as high as conventional biometrics (e.g., face, fingerprints, or iris recognition).

## 5. Discussion

In order to improve the accuracy as much as state-of-the-art biometrics, we performed optimization to derive various feature combinations through finger channel increase and frequency combination search. By increasing the finger channels, we converted multi-channel signals into 2D images, compared them visually, and performed 2D CNN analysis to evaluate and compare performance. In addition, all the accuracies of each combination of finger channels and spectral features were evaluated and compared. We also found that frequency selection optimization not only increased the efficiency of learning and authentication speed but also improved accuracy. Compared with using all frequency features that resulted in an accuracy of 99.08%, the highest accuracy was 99.62% when 47 frequencies (1/3 of the original features) were selected. Our work improved the accuracy of previously reported methods [19] from 95.89% (41 subjects) to 99.62% (54 subjects) and EER from 1.08% to 0.089%. Our proposed method provides promising results comparable to state-of-the-art biometrics approaches.

Our approach holds considerable potential as it analyzes the internal mechanical properties of the fingers and hands. The anatomical constituents of the fingers and hands include numerous cells and muscles, fat, ligaments and cartilage, and multiple layers of tissues, and thus a model of tremendous complexity is required to explain those constituents. In particular, the 3D location-specific signal mapping results of the fingers and hands in our previous study suggest that the anatomical features are reflected, especially for the bones and joints [19]. Based on these data, it can be expected that the obtained features will have permanence under the assumption that the anatomical features, including the bones of our hands, do not change significantly over a certain period.

Another key aspect of our work is that we suggest a method to increase the overall operation speed in comparison with our previous study. Compared with the previous study, we optimized the hardware operation by decreasing the frequency scanning points by half and reducing the acquisition time at each frequency by half. In addition, through frequency feature selection, acquisition frequency features were reduced without loss of accuracy, thus shortening the authentication time of the overall system. Through all these processes, the authentication time could be drastically reduced from 15 s to 2.5 s. By selecting fewer numbers of frequency features, the operation time can be further reduced to less than a second. This could suggest a method to overcome the limitations of acoustic biometric authentication research for commercialization. In addition, through the hardware design of the 1:5 (1 send: 5 receive) method, a system capable of simultaneously acquiring five multi-channel signals from one acoustic selection was implemented. Therefore, despite the further increase in the number of channels and the number of features of the acquired signal, authentication can be performed without slowing down the operation speed.

It can be a reasonable strategy to use our proposed method as a complementary tool for existing biometrics, as previously demonstrated in fingerprint, ECG, iris, heartbeat, and others. The multimodal approach can not only increase the level of security by detecting the liveness of the user attempting to meet the requirements of real-world applications [70] but also improve the accuracy by using score-level or feature-level fusions. Here, our approach provides unique advantages such as (a) inherent liveness detection by employing the characteristics obtained from the live biological tissues, (b) less susceptibility to changes in illumination and pose unlike other biometric traits such as fingerprint, iris, vein, and face, and (c) difficulty to counterfeit because of the need to reproduce the mechanical properties of internal body structures. Despite the unique advantages of our approach, there is a limitation in that the range of efficacy demonstrated through this study may be limited in terms of the demographics of the subjects participating in this study and the total period of data acquisition and evaluation. It may be difficult to guarantee the validity of the results for groups that are not included in the range of race or age groups convened in this study. Additionally, undesirable physiological changes due to variability in body mass index which were not investigated in this study may cause another artifact in the characteristics of biometric features. This study reports the results of analysis only based on data obtained through experiments conducted for up to several months, but if time changes for a period longer than several years may cause fluctuations due to undesired effect on the mechanical characteristics of the body. To address these limitations, it may be necessary to retrain the classification model by methods such as incremental learning that continuously use input data to extend the learning model to reflect temporal variability. In addition, in order to defend against unregistered intruders, it may be necessary to conduct research on anomaly detection that can detect a difference from the secured normal data pattern. Therefore, additional experiments and evaluations may be necessary in the long term along with physiological changes through sufficient datasets obtained from subjects of various ages and races. Additionally, a multimodal approach combined with well-established methods (e.g., iris, fingerprint) can overcome the shortcomings of our suggested method. Future work toward multimodal biometrics may allow comparative studies of the proposed method from different perspectives to be conducted. For example, comparative studies of various other approaches such as classical sum rule, weighted sum rule, and fuzzy logic in each authentication method would be essential for the development of multimodal-based high-performance biometric authentication with other modalities [71].

There are aspects of tradeoff between convenience and high security in biometric authentication. The ideal identity authentication has high security as well as convenience, but in reality, biometric systems either have higher convenience but lower security or vice versa. Therefore, for a wide range of practical applications, we need to choose a biometric system design that compromises between accuracy and convenience. For example, increasing the number of sensors or combining multiple biometric traits may cause not only user inconvenience but also the complexity of the hardware design. Moreover, the tradeoff can be significant depending on applications such as mobile and wearable authentications. This type of applications requires a compact and simple design, so the increasing size, cost, power requirements, sensing channels, and processing logics may be a potential drawback. In contrast, in areas requiring robust security such as financial transactions, higher accuracy is required rather than convenience. To properly quantify the security and convenience characteristics for different biometric applications, a general baseline of how such systems are to be made will be beneficial. In our study, various feature combinations that optimize the authentication process, such as combining finger channels or measurement frequencies, were derived and thoroughly evaluated. For example, in terms of authentication time, accuracy, cost, and convenience according to the number of sensing channels and optimal frequency selection, this study can provide a guidance for designing a well-suited system in a specific application field.

## 6. Conclusions

Bioacoustic-based biometric applications can provide precise and robust identity authentication with accuracies comparable to those of conventional biometric methods. For a broad range of practical applications, such as unobtrusive and continuous identity authentication based on wearable bioacoustics, various feature combinations that optimize the authentication process, such as the combination of finger channels or measurement frequencies, were derived. The classification accuracies according to these design parameters were analyzed with the use of multiple machine learning models and compared with CNN. The experimental results show that a deep CNN algorithm can automatically and adaptively learn and optimize the spatial hierarchies of features, and traditional machine learning algorithms with suitable parameters and well-refined manual feature extraction can perform as well as CNN or better. Accordingly, our study could be used for designing systems that needs to balance system complexity and accuracy. Existing biometric recognition technologies utilize specific parts of the human body (e.g., fingerprint, face, and iris) and require the cognitive action of a user to place the body part on or in front of the biometric sensor. However, this is inconvenient for mobile and wearable devices (e.g., unlocking a mobile phone or logging into a website). In contrast, our technology can be applied to any part of the human body, allowing continuous identity recognition using wearable and mobile devices, and can thus enable seamless human–computer interactions.

## Figures and Tables

**Figure 1 biosensors-12-00700-f001:**
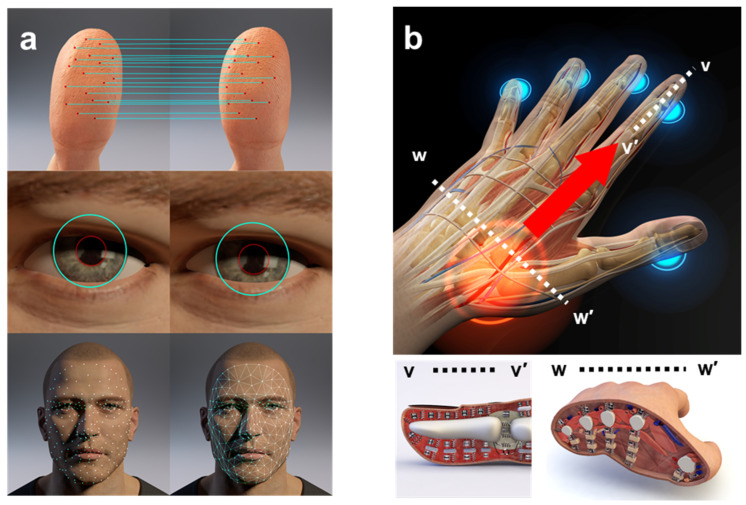
Conceptual diagram of the bioacoustics-based identity authentication approach. (**a**) Conventional biometrics comparing structural similarities between acquired images. (**b**) Proposed biometric authentication method using characteristics inside the body.

**Figure 2 biosensors-12-00700-f002:**
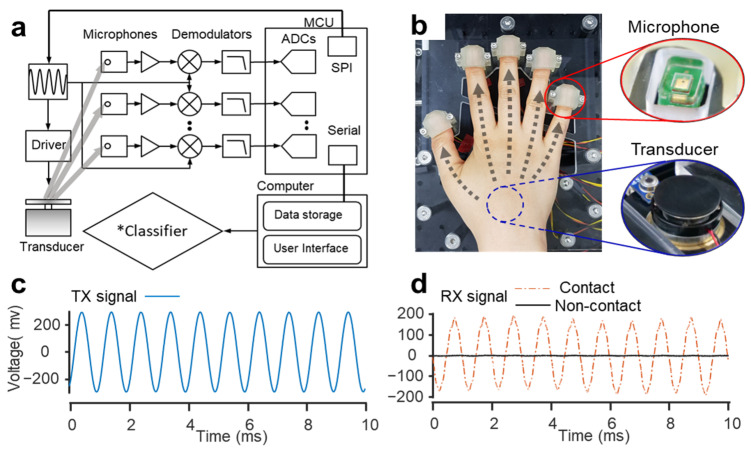
Design of the multichannel bioacoustic identity authentication system. (**a**) Schematic of the acoustic-spectra-based identity authentication system. The acquired data were transmitted to the user identification module. * Classifiers of choice in our proposed system were Random Forest (RF), kNN (k Nearest Neighbor), LDA (Linear Discriminant Analysis), SVM (Support Vector Machine), and CNN (Convolutional Neural Network). (**b**) Photo of the system simultaneously measuring finger acoustic-transmitted spectra from five microphones by placing the hand of the subject on the platform. (**c**) The shape of the acoustic impedance spectrum applied to a finger through a bone conduction transducer. (**d**) The difference in the signal received from the microphone depending on whether the transducer is in contact with the hand.

**Figure 3 biosensors-12-00700-f003:**
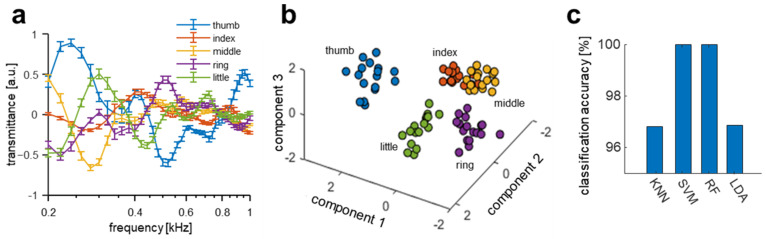
Acoustic spectral variations and classifications between fingers. (**a**) Acoustic spectra of each finger (thumb, index, middle, ring, and little) of a subject. The error bars in the figure indicate the relative standard deviation. (**b**) Scatter plot of the first three principal components of the acoustic spectra for each finger from principal component analysis (PCA). (**c**) Comparison of finger classification accuracy from four different classifiers.

**Figure 4 biosensors-12-00700-f004:**
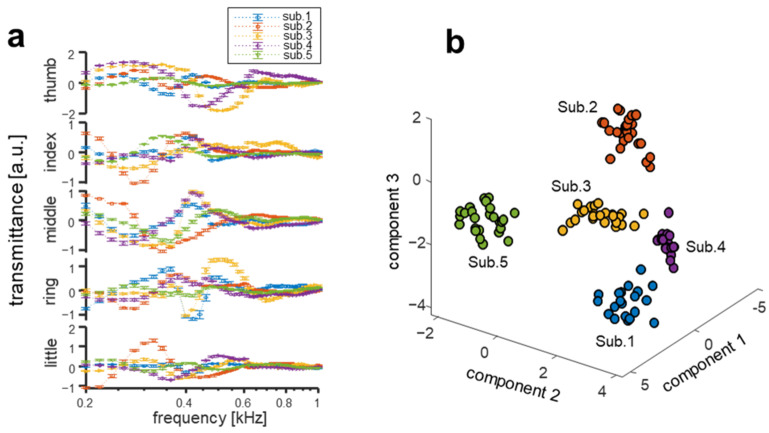
Interpersonal discernibility of multichannel bioacoustics signals. (**a**) Finger acoustic spectra from five individual subjects used to assess interpersonal variation. The error bars in the figure indicate the relative standard deviation. (**b**) Scatter plot of the first three principal components of the acoustic spectra for five subjects from PCA.

**Figure 5 biosensors-12-00700-f005:**
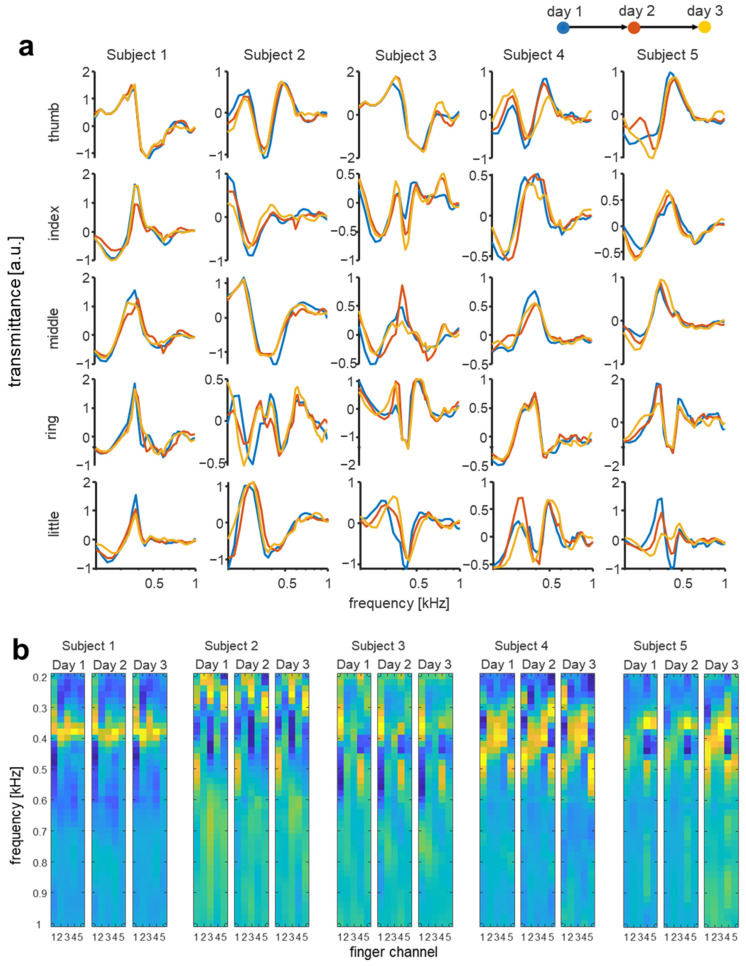
Temporal variation in each finger acoustic spectra in five subjects for three days. (**a**) The acoustic spectra of the fingers were measured at two-week intervals. (**b**) Two-dimensional image intensity of five-finger channel acoustic spectra of five subjects for three days.

**Figure 6 biosensors-12-00700-f006:**
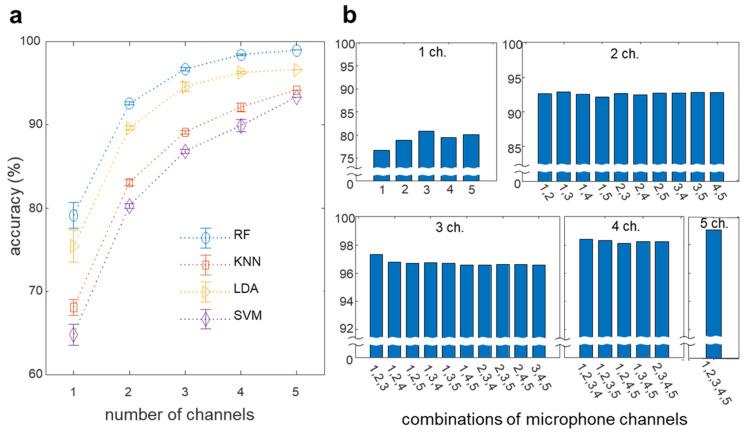
Identity recognition based on multichannel bioacoustics. (**a**) Comparison of identification accuracy for four models according to the number of channels. (**b**) Identification accuracy in the RF according to finger channel combination.

**Figure 7 biosensors-12-00700-f007:**
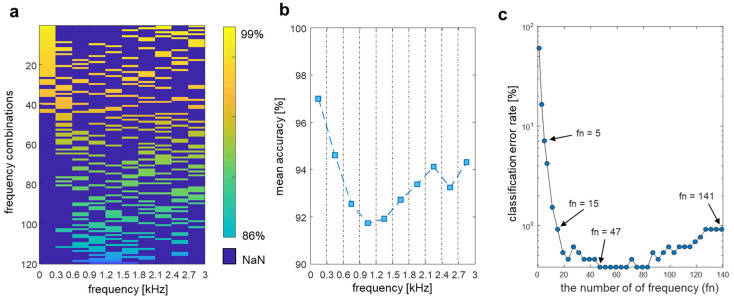
Identity recognition feature optimization using selective frequency scan. (**a**) The accuracy of each selected frequency combination. The more accurate combination is located at higher positions in the image, and lighter yellow indicates higher accuracy. (**b**) Average classification accuracy of each frequency sections in all combinations. (**c**) Classification accuracy with the increasing number of frequencies using the forward-selection method.

**Figure 8 biosensors-12-00700-f008:**
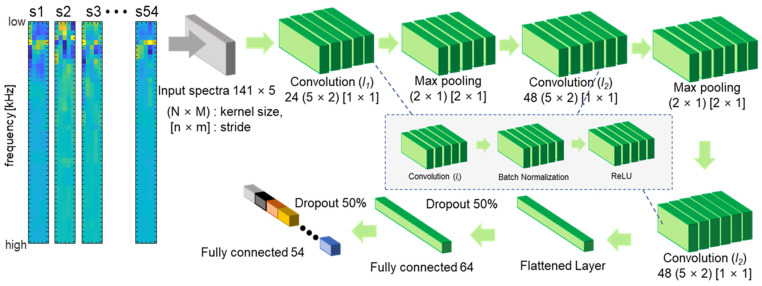
Architecture of the implemented CNN model. Five-channel finger data consisting of 141 samples from 54 subjects were converted into two-dimensional (141 × 5) forms and used as input.

**Figure 9 biosensors-12-00700-f009:**
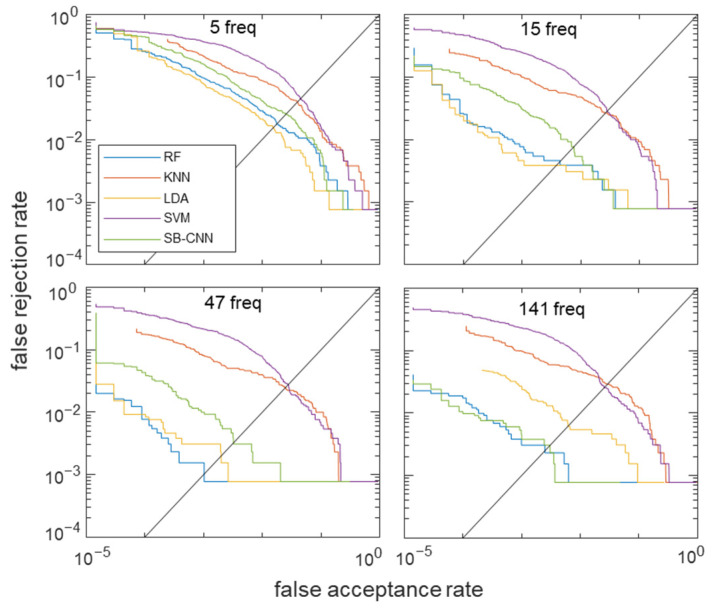
ROC of the acoustic spectra for five classifiers of RF, kNN, LDA, SVM, and SB-CNN for each number of frequencies.

**Figure 10 biosensors-12-00700-f010:**
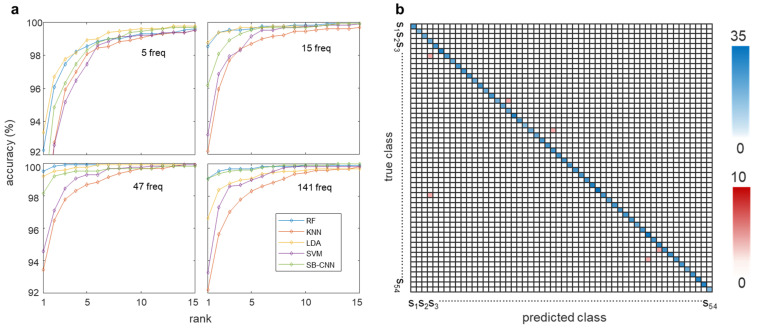
Individual identification accuracy of each classification model according to four different numbers of frequencies (5, 15, 47, and 141). (**a**) The cumulative matching characteristic (CMC) curves for the classifier models for each case of selected frequencies. (**b**) Confusion matrix of the RF classifier among 54 subjects in the case of 47 frequencies.

**Figure 11 biosensors-12-00700-f011:**
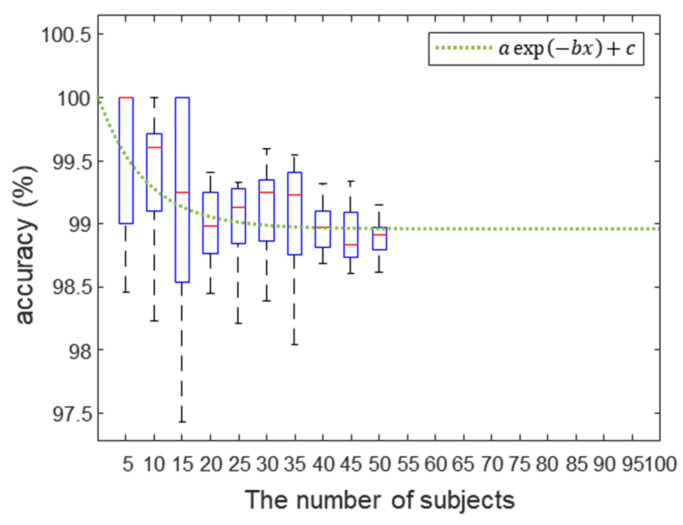
The boxplots of the accuracy according to the number of subjects. Each classification accuracy for the selected dataset. The green dotted line represents the accuracy prediction curve as the number of subjects increases. The inset equation’s parameters were 1.056, 0.1194, and 98.96 for a, b, and c, respectively.

**Table 1 biosensors-12-00700-t001:** Accuracy, EER, and AUC of Discriminative Classification Models According to Four Different Numbers of Frequencies.

	Fn*	Accuracy [%]				EER [%]				AUC			
Classifier		5	15	47	141	5	15	47	141	5	15	47	141
*RF*		92.28	98.55	99.62	99.08	1.7584	0.4479	0.0887	0.2423	0.9975	0.9993	0.9996	0.9998
*kNN*		87.54	92.20	93.43	92.20	3.9755	2.6758	2.4465	2.9052	0.9930	0.9963	0.9967	0.9957
*LDA*		93.35	98.78	99.24	96.64	1.5291	0.3823	0.2135	0.6203	0.9986	0.9996	0.9997	0.9994
*SVM*		85.17	93.20	94.57	93.27	4.5107	2.8901	2.7501	2.5828	0.9908	0.9955	0.9959	0.9956
*SB-CNN*		88.99	96.18	98.17	99.08	2.5871	0.7645	0.3109	0.3058	0.9974	0.9997	0.9997	0.9999

Fn*: The number of frequencies.

**Table 2 biosensors-12-00700-t002:** Processing Time of Train and Prediction for Each Classifier.

Classifier	Processing Time	
	Training (s)	Prediction (ms)
*RF*	2.1114	40.54
*kNN*	0.1028	2.34
*LDA*	0.2623	24.60
*SVM*	17.4162	212.16
*SB-CNN*	2460 (41 min)/2000 epoch	0.16

## Data Availability

The data presented in this study are available from the corresponding author on reasonable request.

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
