# Peer review of "Multichannel Acoustic Spectroscopy of the Human Body for Inviolable Biometric Authentication"

_biosensors, 2022, doi:10.3390/bios12090700_

Round 1
Reviewer 1 Report
First I would like to thank the authors for their submission. The submission is sound and very well motivated and every study design decision has been well accounted for. This paper presents the design and evaluation of a multichannel identity authentication system based on 74 bioacoustics spectroscopy. They analyzed the signals from each finger channel were recorded from five subjects to determine 242 the differences in the signal transmission characteristics of each person. As well as verified the temporal stability of the acoustic transmission signal, the finger acoustic 266 spectra of the participants were acquired on three different days at two-week interval. They achieved an accuracy of 99.62%, which is comparable to the existing biometric methods.
The paper is very well structured and I loud the authors of how complete the presented research is. While reading the first part I was wondering how the size of 5 participants might influence the results, then seeing the scalability study resolved my concerns. Hence, I would recommend accepting this submission. Maybe one minor comment about missing relevant reference:
Sarah Faltaous, Jonathan Liebers; Yomna Abdelrahman, Florian Alt; Schneegass, Stefan: VPID: Towards Vein Pattern Identification Using Thermal Imaging. In: i-com (2019) Nr. 18 (3), S. 259-270. doi:10.1515/icom-2019-0009
Author Response
Dear Referee:
We appreciate the thoughtful guidance from the Reviewer. We are grateful for the detailed comments and excellent suggestions, which have substantially improved the manuscript. We have prepared this revised manuscript closely following your suggestions. In this letter, we outline the changes that were made. Below your comments are in italic, underlined, and blue coloured. Our responses are in regular Times New Roman.
Sincerely,
Joo Yong Sim

Reviewer 2 Report
Authors propose a novel biometrics authentication method by using multichannel accoustic to scan a body part such as a hand.
There are several issues need to be carefully addressed:
1st,as a biometrics, it must be unique and invariant as far time and measurement concerned. By using multichannel accoustic, what structure and features are we measuring? are these as invariant as iris?
2nd,to prove an innovation of biometrics method, the experimental must designed to show its invariant as well as uniqueness. as such, classification accuracy is far not adequate.
Maybe, it can be good idea to be used as complementary tools for existing biometrics techniques.
Author Response

(The authors gave the same response as above.)

Reviewer 3 Report
The study is sound overall. However, further clarification in the method is needed.
1. The authors mentioned the use of modulated signal. However, no description of how the signal was modulated. It seemed that they used the pure sinusoidal signals at different frequencies. Please clarify.
2. In the Figure 2a, what did the the demodulators do?. There was a low pass filter before the ADC. What was the cutoff frequency? What was the sampling frequency of ADC.
3. How was the signal transmission characteristics measured? (line 243)
4. At line 36, what were physical objects? ID cards? please be specific.
5. At line 169, "the transducer can simultaneously receive signal from multiple sensor". Is the statement correct? The transducer was emitting signals and the sensors detect signals. Please clarify.
6. At line 181, should the word "increase" be "decrease"?
7. In figure 3b, the vertical label should be in the same direction as the figure 3a and 3c. Same for figure 4b.
8. At line 415, please rephrase "five-channel data of each finger". Did each finger have one channel of data?
Author Response

(The authors gave the same response as above.)

Round 2
Reviewer 3 Report
The authors have addressed most of the concerns in the first review. Further clarification is still needed.
The authors explained the frequency modulation in the paper. However, it should not be called as frequency modulation, which means the frequency of the signal is controlled by a separate signal. E.g. FM radio signal. Frequency selection is a better term for this study because the signals used in the measurement are single frequency signals. The device can select different frequencies. Please change "frequency modulation" to "frequency selection"
The demodulation is to retrieve the amplitude of the signal. Is the sampling frequency too low? The low pass filter has the cutoff frequency of 20Hz, which means the highest frequency in the signal for ADC could be 20Hz. If so, 18.8 Hz sampling frequency does not meet the requirement of Nyquist law. Please clarify.
Author Response

(The authors gave the same response as above.)
